# Neuron-specific enolase and neuroimaging for prognostication after cardiac arrest treated with targeted temperature management

Soo Hyun Kim[1], Hyo Joon Kim[2], Kyu Nam Park[2], Seung Pill Choi[1], Byung Kook Lee[3], Sang Hoon Oh[2], Kyung Woon Jeung[3], In Soo Cho[4], Chun Song Youn[2]*

1 Department of Emergency Medicine, Eunpyeong St. Mary Hospital, College of Medicine, The Catholic University of Korea, Seoul, South Korea, 2 Department of Emergency Medicine, Seoul St. Mary Hospital, College of Medicine, The Catholic University of Korea, Seoul, South Korea, 3 Department of Emergency Medicine, Chonnam National University Medical School, Gwangju, Korea, 4 Department of Emergency Medicine, Hanil General Hospital, Korea Electric Power Medical Corporation, Seoul, Korea

* ycs1005@catholic.ac.kr

## Abstract

### Background

Prognostication after cardiac arrest (CA) needs a multimodal approach, but the optimal method is not known. We tested the hypothesis that the combination of neuron-specific enolase (NSE) and neuroimaging could improve outcome prediction after CA treated with targeted temperature management (TTM).

### Methods

A retrospective observational cohort study was performed on patients who underwent at least one NSE measurement between 48 and 72 hr; received both a brain computed tomography (CT) scan within 24 hr and diffusion-weighted magnetic resonance imaging (DW-MRI) within 7 days after return of spontaneous circulation (ROSC); and were treated with TTM after out-of-hospital CA between 2009 and 2017 at the Seoul St. Mary's Hospital in Korea. The primary outcome was a poor neurological outcome at 6 months after CA, defined as a cerebral performance category of 3–5.

### Results

A total of 109 subjects underwent all three tests and were ultimately included in this study. Thirty-four subjects (31.2%) experienced good neurological outcomes at 6 months after CA. The gray matter to white matter attenuation ratio (GWR) was weakly correlated with the mean apparent diffusion coefficient (ADC), PV400 and NSE (Spearman's rho: 0.359, -0.362 and -0.263, respectively). NSE was strongly correlated with the mean ADC and PV400 (Spearman's rho: -0.623 and 0.666, respectively). Serum NSE had the highest predictive value among the single parameters (area under the curve (AUC) 0.912, sensitivity 70.7% for maintaining 100% specificity). The combination of a DWI parameter (mean ADC or PV400)

**Data Availability Statement:** All relevant data are within the manuscript and its Supporting Information files.

**Funding:** This research was supported by Basic Science Research Program through the National Research Foundation of Korea (NRF) funded by the Ministry of Education (2018R1D1A1B07047594)

**Competing interests:** The authors have declared that no competing interests exist.

and NSE had better prognostic performance than the combination of the CT parameter (GWR) and NSE. The addition of the GWR to a DWI parameter and NSE did not improve the prediction of neurological outcomes.

## Conclusion

The GWR ($\leq$ 24 hr) is weakly correlated with the mean ADC ($\leq$ 7 days) and NSE (highest between 48 and 72 hr). The combination of a DWI parameter and NSE has better prognostic performance than the combination of the GWR and NSE. The addition of the GWR to a DWI parameter and NSE does not improve the prediction of neurological outcomes after CA treatment with TTM.

## Introduction

The prediction of neurological outcomes after cardiac arrest (CA) is challenging and critically important [1]. Targeted temperature management (TTM) between 32 and 36˚C for at least 24 hr, for both shockable and nonshockable rhythms, is generally considered standard care after CA. Even though TTM can improve neurological outcomes and survival [2–8], two-thirds of patients will die after CA due to hypoxic-ischemic brain injury [9]. However, the leading cause of death is the withdrawal of life-sustaining treatment (WLST) because of failure to awaken [10–12]. Therefore, special precautions should be taken to avoid false outcome predictions, which affect the decisions regarding WLST [9].

Unimodal approaches, using clinical examinations, neuron-specific enolase (NSE), somato-sensory evoked potential (SSEP), electroencephalography (EEG), brain computed tomography (CT) and diffusion-weighted magnetic resonance imaging (DW-MRI), have been tested as predictors of neurological outcomes [13–23]. However, no single test can predict poor neuro-logical outcomes with a 0% false-positive rate (FPR) in patients treated with TTM. Therefore, the combination of multiple modalities is recommended to avoid premature WLST and mini-mize the risk of an erroneous prognostication of poor outcomes [24, 25]. Several multimodal prognostication studies have been reported using different prognostic tools [26–28]; however, the ideal combination is unknown.

We have previously reported that brain CT and brain DW-MRI can predict poor neurologi-cal outcomes in patients treated with TTM after CA [21, 29]. The purpose of this study was to determine whether combinations of multiple tests would perform better than individual tests in patients treated with TTM after CA. The combination of NSE and neuroimaging, which was used in the largest number of patients in our hospital, was an ideal model for testing this hypothesis. We hypothesized that the combination of NSE and neuroimaging could improve outcome prediction after CA treated with TTM. We performed a retrospective analysis to test whether the combination of NSE, a quantitative analysis of brain CT calculated by the gray matter to white matter attenuation ratio (GWR) and a quantitative analysis of brain DW-MRI could improve diagnostic performance for predicting outcomes after CA.

## Methods

### Study design and setting

This retrospective observational study using prospectively collected data was performed at Seoul St. Mary's Hospital in Seoul, Korea, between January 2009 and December 2017. This

study was approved by the Institutional Review Board of Seoul St. Mary's Hospital; waiver of consent was allowed because of the retrospective nature of the study. We included patients who underwent at least one NSE value measurement between 48 and 72 hr after return of spontaneous circulation (ROSC) and received both a brain CT scan within 24 hr after ROSC and DW-MRI within 7 days after ROSC. The exclusion criteria were as follows: age<18 yr, CA due to trauma or intracranial hemorrhage, a previous history of neurological disease and CT or DW-MRI with a poor image quality.

All patients who were comatose after ROSC were treated according to our previously published post-CA care protocols [30]. The induction and maintenance of TTM were performed using an endovascular cooling device (Thermoguard, ZOLL Medical Corporation, Chelmsford, MA, USA) or ArticSun (Bard Medical, Louisville, CO, USA) to obtain a target temperature of 33°C, which was maintained for 24 hr. Sedative (midazolam) and paralyzing (rocuronium) agents were immediately administered to control shivering before induction, followed by continuous infusion during the entire TTM phase. After 24 hr at 33°C, the patients were slowly rewarmed to 37°C at a rate of 0.25°C/h.

## NSE

Blood samples were obtained at 48 and 72 hr after ROSC. We used the highest NSE values obtained between 48 and 72 hr after ROSC in this study because they are known to have similar predictive values for predicting poor neurological outcomes [15]. In the majority of the patients, at least one sample was collected between 48 and 72 hr after ROSC, which minimized the effect of missing data from a single time point.

## CT

Brain CT scans were obtained on a 64-channel scanner (Light Speed VCT scanner; GE Healthcare, Milwaukee, Wisconsin, USA) with 5 mm slices. The GWR was calculated by a blinded investigator as previously reported [21]. Briefly, Hounsfield units (HU) were recorded in the caudate nucleus, putamen, genu of corpus callosum, and posterior limb of the internal capsule at the basal ganglia level, as well as the medial cortex and medial white matter at the cerebrum and high convexity area. The average gray matter to white matter ratio (aGWR), which was defined as the mean of the basal ganglia GWR and cerebrum GWR, was used in the current analysis. Patients were divided into three groups according to their aGWR: severe edema (aGWR < 1.1), mild edema (aGWR 1.1–1.2) and no edema (aGWR ≥ 1.2).

## DWI

Brain DWI-MR images obtained within 7 days after ROSC were analyzed. The imaging protocol and analysis were the same as those published previously [30]. Briefly, the standard of b = 1000 s/mm$^2$ was used for all DW images, and apparent diffusion coefficient (ADC) maps were created with commercial software and a workstation system (Leonardo MR Workplace; Siemens Medical Solutions, Erlangen, Germany). The images were processed and analyzed on a PC using the FMRIB Software Library (Release 5.0 (c) 2012, The University of Oxford), which can extract brain tissue images from the skull and optic structures and measure the mean ADC of the extracted brain image or the absolute volume of voxels that have a predefined ADC range. The ADC is a measure of the diffusion coefficient of water molecules within tissue and is calculated using MRI with DW imaging. The images were retrieved in Digital Imaging and Communications in Medicine (DICOM) format from picture archiving and communication system (PACS) servers at the hospital and converted into NITFI format using the program MRIcron (http://www.nitrc.org/projects/mricron). To exclude artifacts or

cerebrospinal fluid, voxels with ADC values below $50^{-6}$ mm$^2$/s or above $1200^{-6}$ mm$^2$/s were excluded from the analysis. The % voxels of a low ADC value (PV), meaning the percentage of voxels below different ADC thresholds, was calculated. Previous studies have shown that PV400, which represents the % voxel below the ADC value of $400 \times 10^{-6}$ mm$^2$/s, is strongly associated with neurological outcomes after CA; therefore, this value was used in the current study [30].

## Outcome assessment

The outcome of this study was a poor neurological outcome at 6 months after CA, defined as a cerebral performance category (CPC) between 3 and 5. Follow-up was performed either face to face or via the telephone. The CPC scale ranges from 1 to 5: 1 represents good cerebral performance or slight cerebral disability, 2 represents moderate disability or independent activities of daily life, 3 represents severe disability or dependence on others for daily support, 4 represents a comatose or vegetative state, and 5 represents death or brain death.

## Statistical analysis

Continuous data are presented as the median and interquartile range (IQR) when nonnormally distributed, and categorical variables are presented as counts and percentages. To compare differences in patient characteristics and outcomes, chi-square tests or Fisher's exact tests were performed for categorical variables, and Mann-Whitney U tests were performed for continuous variables.

To evaluate and compare the equivalence of the area under the curve (AUC) using the Delong test, the predictive performance was determined using a receiving operating characteristic (ROC) curve established with logistic regression models. First, we determined the AUC of the following single parameters using ROC curves: GWR, mean ADC, PV400 and highest NSE level. Then, the AUC was obtained by combining two single tests. Finally, the AUC was calculated by combining three single tests (i.e., GWR + mean ADC + highest NSE value and GWR + PV400 + highest NSE value). To evaluate the prognostic values of a single test and a combination of multiple tests, sensitivities, specificities, positive predictive values and negative predictive values, as well as their 95% confidence intervals, were calculated. Statistical analyses were performed using SPSS 20.0 (Chicago, IL) and MedCalc 15.2.2 (MedCalc Software, Mariakerke, Belgium). P values $\leq 0.05$ were considered statistically significant.

## Results

### Patient demographics

Of the total of 355 subjects who underwent TTM during the study period, 246 were excluded owing to the lack of CT imaging (n = 85), a poor imaging quality (n = 28), death within 48 hr (n = 39), the lack of DWI imaging, due to the alert, hemodynamically unstable, refused treatment or brain death (n = 79), or the lack of NSE (n = 15). All three tests were performed in 109 subjects who were ultimately included in this study (Fig 1). The median age was 53 (IQR 40.5, 60.5) years, and 74 (67.9%) subjects were male (Table 1). Ventricular fibrillation was the initial rhythm for 37 (33.9%) subjects. A pulseless electrical activity was observed in 23 (21.1%) subjects; an asystole was recorded in 45 (41.3%) subjects; and the initial rhythm was unknown in the remaining 4 (3.7%) subjects. The cardiac cause of arrest was diagnosed in 63 (57.8%) subjects. The rest had a respiratory cause of arrest (34 subjects, 31.2%), other medical causes (10 subjects, 9.2%), submersion (1 subject, 0.9%), drug intoxication (1 subject), and an unknown

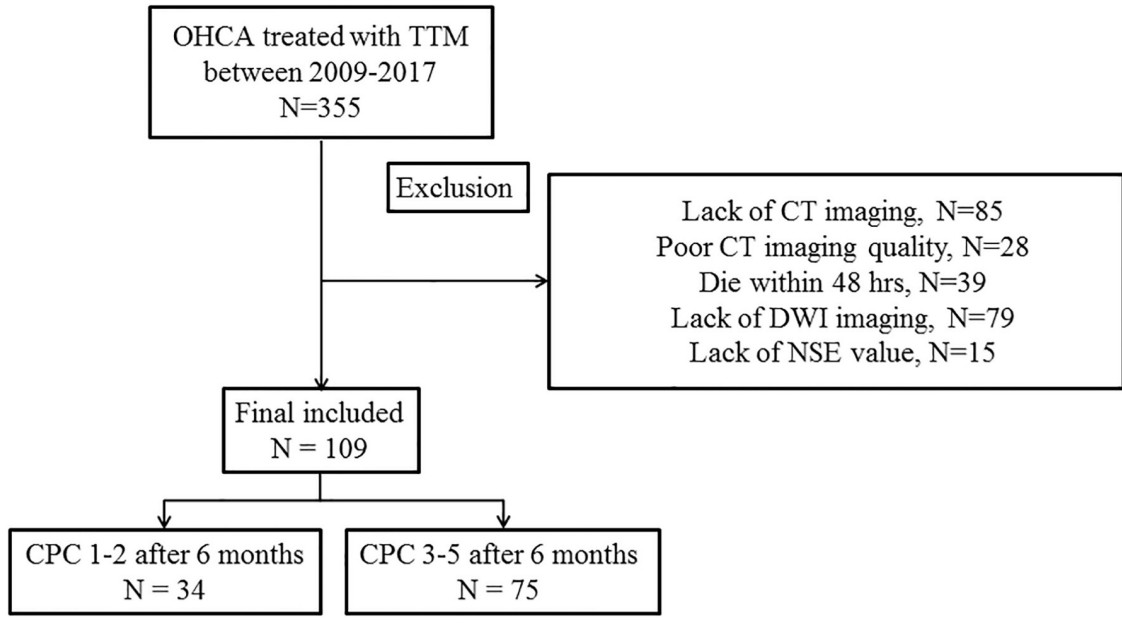

**Fig 1. Flowchart for included patients.**

**Table 1. Baseline characteristics of subjects.**

|  | Good | Poor | P |
|---|---|---|---|
|  | N = 34 | N = 75 |  |
| Age (year) | 53 (40.8–62.3) | 53 (40.0–66.0) | 0.799 |
| Sex, male | 25 (73.5) | 49 (65.3) | 0.508 |
| Past history |  |  |  |
| Healthy | 22 (64.7) | 43 (575.3) | 0.531 |
| HTN | 9 (26.5) | 18 (24.0) | 0.813 |
| DM | 3 (8.8) | 17 (22.7) | 0.111 |
| Witness | 27 (79.4) | 38 (50.7) | 0.006 |
| Bystander CPR | 19 (55.9) | 39 (52.0) | 0.836 |
| VT/VF | 22 (64.7) | 15 (20.0) | <0.001 |
| Cardiac cause of arrest | 31 (91.2) | 32 (42.7) | <0.001 |
| Time from collapse to ROSC, min | 20.0 (10.8–30.0) | 36.0 (26.0–47.0) | <0.001 |
| Brain CT |  |  |  |
| Scan time, min | 23.0 (13.5–36.8) | 22.0 (14.0–33.0) | 0.896 |
| GW ratio | 1.23 (1.19–1.28) | 1.20 (1.16–1.26) | 0.046 |
| Brain DWI |  |  |  |
| Scan time, hr | 73.0 (60.8–80.0) | 68.0 (54.0–81.0) | 0.391 |
| Whole brain ADC, $^{*}$ $10^{-6}$ mm$^2$/s | 798.3 (769.6–833.1) | 653.4 (554.1–735.9) | <0.001 |
| % Brain volume < 400 | 1.10 (0.44–2.44) | 8.77 (2.60–20.86) | <0.001 |
| Highest NSE, μg/L | 23.0 (16.0–36.1) | 102.0 (56.4–186.0) | <0.001 |
| LOS, day | 15 (9–22) | 10 (5–20) | 0.019 |

Data are expressed median (interquartile range) or percentage.

Abbreviation: HTN = Hypertension, DM = Diabetes, CPR = Cardiopulmonary resuscitation, ROSC = return of spontaneous circulation, CT = computed tomography, GW = gray to white matter, DWI = diffusion weighted imaging, ADC = apparent diffusion coefficient. NSE = neuron specific enolase, LOS = length of stay.

cause (1 subject). Thirty-four subjects (31.2%) experienced good neurological outcomes at 6 months after CA.

The median brain CT scan time was 23 min for the good outcome group and 22 min for the poor outcome group. The GWR was significantly lower in the poor outcome group (median 1.20, IQR 1.16–1.26) than in the good outcome group (median 1.23, IQR 1.19–1.28). The median brain DWI scan time was 73 hr (IQR 60.8–80.0) for the good outcome group and 68 hr (IQR 54.0–81.0) for the poor outcome group. The mean ADC was higher in the good outcome group (median $798.3 \times 10^{-6}$ mm$^2$/s; IQR 769.6–833.1) than in the poor outcome group (median $653.4 \times 10^{-6}$ mm$^2$/s; IQR 554.1–735.9). PV400 was lower in the good outcome group (median 1.1%; IQR 0.44–2.44) than in the poor outcome group (median 8.77%; IQR 2.6–20.86). The cutoff values for predicting neurological outcomes at 6 months after CA for the GWR, mean ADC, PV400 and NSE, with 100% specificity, were 1.12, $610 \times 10^{-6}$ mm$^2$/s, 6.5%, and 62.64 µg/L, respectively.

## Association between the GWR and other predictors

The average GWR was weakly correlated with NSE, the mean ADC and PV400 (Spearman's rho = 0.359 (p<0.001), -0.362 (p<0.001), and -0.263 (p = 0.006), respectively) (Fig 2). Among the subjects with a GWR < 1.1, all (7) had an NSE level above 62.64 µg/L, a mean ADC below $610 \times 10^{-6}$ mm$^2$/s and a PV400 above 6.5% (Table 2). However, among the subjects with a GWR ≥ 1.2, 24 (36.9%) had an NSE level above 62.64 µg/L, 54 (81.8%) had a mean ADC above $610 \times 10^{-6}$ mm$^2$/s, and 42 (63.6%) had a PV400 below 6.5%.

## Association between NSE and DWI parameters

The NSE level was strongly correlated with the mean ADC and PV400 (Spearman's rho = -0.623 (p<0.001), and 0.666 (p<0.001), respectively) (Fig 2). Among the subjects with an NSE

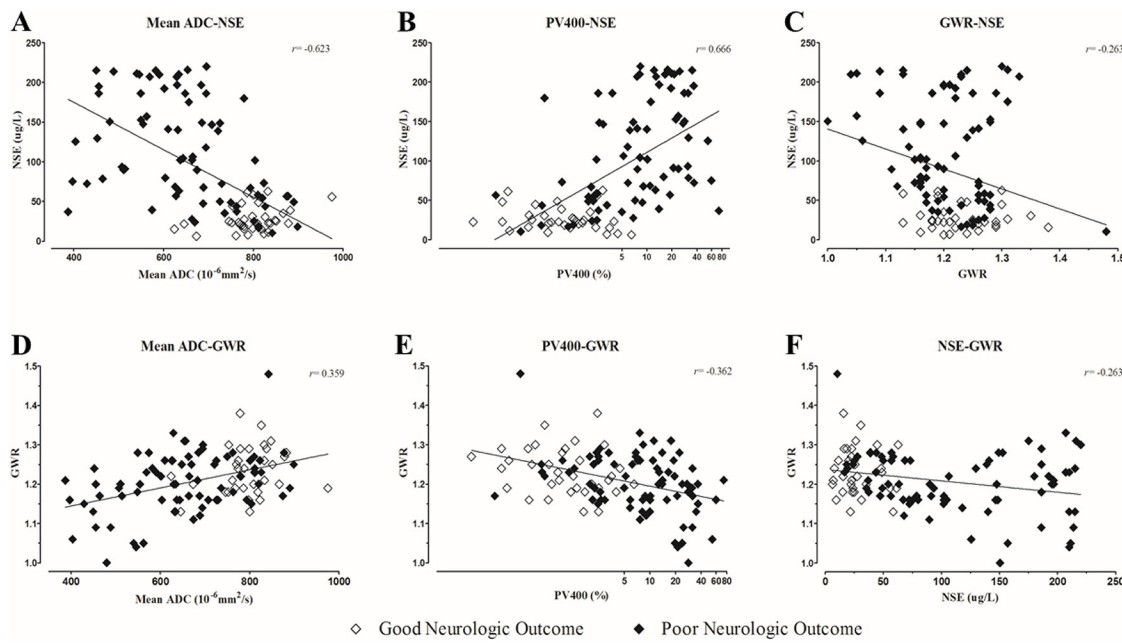

**Fig 2. Correlations between NSE and imaging parameters using Spearman's correlation coefficient.** A: Mean ADC-NSE correlation coefficient *r* = -0.623. B: PV400-NSE correlation coefficient *r* = -0.666. C: GWR-NSE correlation coefficient *r* = -0.263. D: Mean ADC-GWR correlation coefficient *r* = 0.359. E: PV400-GWR correlation coefficient *r* = -0.362. F: NSE-GWR correlation coefficient *r* = -0.263.

**Table 2. Association between GW ratio and other predictors.**

| | Severe Edema | Mild Edema | No Edema |
|---|---|---|---|
| | GW ratio $< 1.1$ | GW ratio 1.1–1.2 | GW ratio $\geq 1.2$ |
| | N = 7 | N = 36 | N = 66 |
| NSE $> 62.64$ μg/L | 7 (100%) | 22 (59.5%) | 24 (36.9%) |
| NSE $\leq 62.64$ μg/L | 0 (0%) | 15 (40.5%) | 41 (63.1%) |
| Mean ADC $\leq 610 * 10^{-6}$ mm²/s | 7 (100%) | 8 (22.2%) | 12 (18.2%) |
| Mean ADC $> 610 * 10^{-6}$ mm²/s | 0 (0%) | 28 (77.8%) | 54 (81.8%) |
| PV400 $> 6.5\%$ | 7 (100%) | 16 (44.4%) | 24 (36.4%) |
| PV400 $\leq 6.5\%$ | 0 (0%) | 20 (55.6%) | 42 (63.6%) |

Abbreviation: GW = gray to white matter, NSE = neuron specific enolase, ADC = apparent diffusion coefficient.

level $> 62.64$ μg/L, 25 (47.2%) had a mean ADC below $610 \times 10^{-6}$ mm²/s, and 41 (77.4%) had a PV400 above 6.5% (Table 3). Among the subjects with an NSE level $\leq 62.64$ μg/L, 54 (96.4%) had a mean ADC above $610 \times 10^{-6}$ mm²/s, and 50 (89.3%) had a PV400 below 6.5%.

## Prognostic value of a single modality

Table 4 shows the areas under the ROC curves of each single test. NSE had the highest predictive value among the single tests (AUC 0.912, sensitivity 70.7% for maintaining 100% specificity). The sensitivities of the GWR, mean ADC, PV400 and NSE for predicting neurological outcomes after CA, with 100% specificity, were 12%, 36%, 62.7%, and 70.7%, respectively.

## Prognostic value of combined modalities

Table 4 shows the areas under the ROC curves for single and multiple tests. The AUC increased when the tests were combined. Of the two test combinations, the AUC (0.929, 95% CI 0.880–0.978) was the highest when PV400 and NSE were combined. Of the three test combinations, the AUC (0.936, 95% CI 0.892–0.980) was the highest when the GWR, PV400 and NSE were combined.

The sensitivity of the mean ADC and NSE combination for predicting neurological outcomes after CA, with 100% specificity, was 73.3%. The sensitivity of the PV400 and NSE combination for predicting neurological outcomes after CA, with 100% specificity, was 78.7%. The addition of the GWR to a DWI parameter and NSE did not improve the sensitivity for the prediction of neurological outcomes (Table 5).

**Table 3. Association between NSE and DWI parameters.**

| | NSE $> 62.64$ μg/L | NSE $\leq 62.64$ μg/L |
|---|---|---|
| | N = 53 | N = 56 |
| Mean ADC $\leq 610 * 10^{-6}$ mm²/s | 25 (47.2%) | 2 (3.6%) |
| Mean ADC $> 610 * 10^{-6}$ mm²/s | 28 (52.8%) | 54 (96.4%) |
| PV400 $> 6.5\%$ | 41 (77.4%) | 6 (10.7%) |
| PV400 $\leq 6.5\%$ | 12 (22.6%) | 50 (89.3%) |

Abbreviation: GW = gray to white matter, NSE = neuron specific enolase, ADC = apparent diffusion coefficient.

**Table 4. Areas under the receiving operator characteristic curves of single and several models combining different outcome predictors.**

| Variable | ROC area | 95% CI |
|---|---|---|
| Single predictor | | |
| GWR | 0.619 | 0.510–0.729 |
| Mean ADC | 0.830 | 0.753–0.907 |
| PV400 | 0.882 | 0.820–0.943 |
| NSE | 0.912 | 0.860–0.964 |
| Combining two predictors | | |
| GWR+Mean ADC | 0.831 | 0.755–0.908 |
| GWR+PV400 | 0.887 | 0.827–0.947 |
| GWR+NSE | 0.918 | 0.868–0.968 |
| Mean ADC+NSE | 0.919 | 0.866–0.972 |
| PV400+NSE | 0.929 | 0.880–0.978 |
| Combining three predictors | | |
| GWR+Mean ADC+NSE | 0.932 | 0.885–0.978 |
| GWR+PV400+NSE | 0.936 | 0.892–0.980 |

Abbreviation: GWR = gray to white matter ratio, ADC = apparent diffusion coefficient, NSE = neuron specific enolase.

**Table 5. Sensitivity, specificity, PPV, NPV for predicting neurological outcome at 6 months after CA of single and multimodal approach.**

| | Cutoff | Poor | Good | Sensitivity | Specificity | PPV | NPV |
|---|---|---|---|---|---|---|---|
| Single | | | | | | | |
| GWR | ≤ 1.12 | 9 | 0 | 12 | 100 | 100 | 34 |
| | | 66 | 34 | (5.6–21.6) | (89.7–100) | (63.1–100) | (24.8–44.2) |
| ADC | ≤ 610 | 27 | | 36 | 100 | 100 | 41.5 |
| | | 48 | 34 | (25.2–47.9) | (89.7–100) | (87.2–100) | (30.7–52.9) |
| PV400 | > 6.5 | 47 | 0 | 62.7 | 100 | 100 | 54.8 |
| | | 28 | 34 | (50.7–73.6) | (89.7–100) | (92.5–100) | (41.6–67.6) |
| NSE | > 62.64 | 53 | 0 | 70.7 | 100 | 100 | 60.7 |
| | | 22 | 34 | (59.0–80.6) | (89.7–100) | (93.3–100) | (46.6–73.6) |
| Multimodal Combination of two tests | | | | | | | |
| GWR or mean ADC | | 29 | 0 | 38.7 | 100 | 100 | 42.5 |
| | | 46 | 34 | (27.6–50.6) | (89.7–100) | | (38.2–46.9) |
| GWR or PV400 | | 47 | 0 | 62.7 | 100 | 100 | 54.8 |
| | | 28 | 34 | (50.7–73.6) | (89.7–100) | | (47.5–62.0) |
| GWR or NSE | | 53 | 0 | 70.7 | 100 | 100 | 60.7 |
| | | 22 | 34 | (59.0–80.6) | (89.7–100) | | (46.6–73.6) |
| Mean ADC or NSE | | 55 | 0 | 73.3 | 100 | 100 | 63.0 |
| | | 20 | 34 | (61.9–82.9) | (89.7–100) | | (53.9–71.2) |
| PV400 or NSE | | 59 | 0 | 78.7 | 100 | 100 | 68.0 |
| | | 16 | 34 | (67.7–87.3) | (89.7–100) | | (57.9–76.7) |
| Combination of three tests | | | | | | | |
| GWR or mean ADC or NSE | | 55 | 0 | 73.3 | 100 | 100 | 63.0 |
| | | 20 | 34 | (61.9–82.9) | (89.7–100) | | (53.9–71.2) |
| GWR or PV400 or NSE | | 59 | 0 | 78.7 | 100 | 100 | 68.0 |
| | | 16 | 34 | (67.7–87.3) | (89.7–100) | | (57.9–76.7) |

Abbreviation: GWR = gray to white matter ratio, ADC = apparent diffusion coefficient, NSE = neuron specific enolase.

## Discussion

The main findings of this study are as follows: 1) the GWR is weakly correlated with the mean ADC, PV400 and NSE, whereas NSE is strongly correlated with the mean ADC and PV400; 2) serum NSE has the highest predictive value among the single tests (AUC 0.912, sensitivity 70.7% for maintaining 100% specificity); 3) the combination of a DWI parameter (mean ADC or PV400) and NSE has better prognostic performance than the combination of the CT parameter (GWR) and NSE; and 4) the addition of the GWR to a DWI parameter and NSE does not improve the prediction of neurological outcomes.

The multimodal approach is recommended to avoid premature WLST and minimize the risk of erroneous prognostication of a poor outcome because no single test so far can predict neurological outcomes with a 0% FPR [24, 25]. Several multimodal approach studies have been reported with different combinations of prognostic tools, but the ideal combination is not known [26–28]. Importantly, not all tests are required for all patients, and all tests cannot be performed for all patients. The guidelines from the European Resuscitation Council and European Society of Intensive Care Medicine suggested a prognostication algorithm for outcome prediction after CA [24]. However, this algorithm should be tested further.

Brain CT is one of the most commonly used tools and has been widely evaluated for prognostication [20–22, 31]. The advantage of CT is that it is relatively easy and safe to perform even in comatose patients, and contrast is not needed in most cases. Brain CT should be performed before TTM to detect intracranial abnormalities. However, brain CT performed at an early stage (especially within 2 hr) has the disadvantage of a low sensitivity [20–22]. When brain CT is repeated, the predictive value increases [32]. Therefore, when discussing the predictive power of brain CT for outcome prediction after CA, the scan time is a very important issue. In our study, brain CT was mostly performed within 2 hr after ROSC. Therefore, the predictive value was relatively low compared with that of other prognostic tools (AUC of a GWR of 0.619; AUCs of the mean ADC, PV400 and NSE of 0.830, 0.882 and 0.912, respectively). Furthermore, the addition of the GWR to a DWI parameter and NSE did not improve the prediction of neurological outcomes after CA treatment with TTM. We do not suggest that brain CT should not be used because it does not help with outcome predictions. This should be limited to brain CT performed early (within 2 hr).

In our study, the sensitivity of the GWR for maintaining 100% specificity was 12%, consistent with the results of previous studies. Lee et al. showed that the combination of brain CT (median scan time 69.5 min) and serum NSE at 48 hr after ROSC had a better prognostic performance than did either test alone in predicting poor neurological outcomes after CA treated with TTM, which is contrary to our study results [33]. The difference between these studies was that Lee et al. used NSE at 48 hr after ROSC, but it is not clear whether there was a difference in results. Further research is needed.

Brain DWI can reveal the structure in detail but has the disadvantage that it is difficult to perform in critically ill patients. Similar to CT, the timing of the examination is important because it is highly likely that no structural damage will be observed in the early stages [23, 34]. The whole-brain ADC was suggested by Wu et al. as an important variable for predicting the neurological prognosis, and in their study, a cutoff value of $665 \times 10^{-6}$ mm$^2$/s for maintaining 100% specificity was suggested, which is slightly higher than the result of our study [35]. Wijman et al. suggested that the percentage of brain volume with an ADC value below 400 to $450 * 10^{-6}$ mm$^2$/s could differentiate between survival with independent and impaired functions [34]. Similar to this study, our previous study showed that PV400 was the best parameter for determining neurological prognosis [29]. However, some researchers have suggested that a

cutoff value below $650 \times 10^{-6}$ mm$^2$/s is good for predicting neurological outcomes after CA; thus, further research is needed [36].

NSE is the most widely studied measure for predicting neurological outcomes after CA and is most commonly used in clinical practice. The best time point for measuring NSE for outcome prediction is 48 or 72 hr after CA [15]. However, NSE has a potential disadvantage called hemolysis because NSE is normally found in red blood cells and platelets. To overcome this issue, care should be taken when measuring NSE, and serial testing is recommended. One prespecified post hoc analysis of the TTM trial showed that the combination of head CT and the highest NSE level at 48 and 72 hr had a higher sensitivity and specificity to predict poor outcomes than CT alone, with no false positives [37]. However, in our study, the addition of the GWR to a DWI parameter and NSE did not improve the prediction of neurological outcomes. Therefore, the additional value of CT in the prognostic model needs further study.

Biomarkers have several advantages. They can be measured at the bedside, are not affected by sedatives and provide a numeric result with little space to interpret. Higher brain biomarker levels indicate more damage to neurons. However, they cannot indicate which part of the brain is most affected. Recently, novel biomarkers, such as serum tau and serum neurofilament light chain (NFL), have been studied, and some data have been published. In a substudy of a TTM trial, Mattsson et al. stated that higher serum levels of tau at 24, 48 and 72 hr after CA correlated with poor neurological outcomes after CA, and serum tau more accurately predicted a poor outcome than did serum NSE [38]. In another substudy of the TTM trial, Moseby-Knappe et al. stated that serum NFL levels at 24, 48 and 72 hr after CA were highly predictive markers of a long-term poor neurological outcome after CA [39]. Both studies used a single molecule array (SIMOA) assay, which is able to accurately quantify proteins or peptides present at low femtomolar concentrations, which are difficult to quantify using conventional techniques. However, SIMOA assays are not yet available for clinical use in most hospitals.

This study has several limitations. First, this was a single-center retrospective study, limiting generalizability. Second, only 109 of the 355 patients who received TTM were included in the final analysis, which could cause selection bias. Therefore, the results of this study may not be generalizable to all CA patients. Third, only patients treated at 33˚C were included in this study. Therefore, our findings cannot be extrapolated to patients who were treated with other TTM temperature targets. Fourth, there may be some intermachine or interprotocol (MRI or FSL program) bias. We did not perform anatomical analyses. For example, the cortex and deep gray nuclei may show different results. Fifth, NSE has the following disadvantages. The NSE threshold is dependent on the timing of the measurement, reflecting the kinetics after the initial release. An additional cause of discrepancies is the variability of the technology used to measure biomarkers, which can lead to significant systematic errors between technologies [40, 41]. Sixth, basal ganglia seem to be the best region to calculate the GWR for predicting a neurological outcome after CA [42]. The AUC of the basal ganglia GWR for predicting the neurological outcomes of our cohort was higher than that of the cerebrum GWR and average GWR (0.625 vs. 0.599 and 0.619, respectively). However, the difference was not statistically significant (basal ganglia GWR vs. average GWR, p = 0.8286). Therefore, using the basal ganglia GWR instead of the average GWR would not change the results of this study. Seventh, the results of these tests were not blinded to the treating physician, potentially creating a self-fulfilling prophecy. Finally, we did not test other prognostic modalities, such as neurological examinations, SSEP, and EEG. Future studies should be conducted to determine the prognostic value of these modalities.

## Conclusion

The GWR ($\leq$ 24 hr) is weakly correlated with the mean ADC ($\leq$ 7 days) and NSE (highest between 48 and 72 hr). The combination of a DWI parameter and NSE has better prognostic performance than the combination of the GWR and NSE. The addition of the GWR to a DWI parameter and NSE does not improve the prediction of neurological outcomes after CA treatment with TTM.

## Supporting information

**S1 Data. Combination.**
(XLSX)

## Author Contributions

**Conceptualization:** Soo Hyun Kim, Chun Song Youn.

**Data curation:** Hyo Joon Kim, Sang Hoon Oh.

**Formal analysis:** Soo Hyun Kim, Hyo Joon Kim, Kyu Nam Park, Seung Pill Choi, Byung Kook Lee, Sang Hoon Oh, Kyung Woon Jeung, In Soo Cho, Chun Song Youn.

**Funding acquisition:** Chun Song Youn.

**Writing – original draft:** Soo Hyun Kim, Kyu Nam Park, Chun Song Youn.

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
