## [Decision Letter · Decision Letter 0]

2 Jun 2020

PONE-D-20-14003

Correlation between neuron-specific enolase and neuroimaging and their prognostic utility after cardiac arrest treated with targeted temperature management

PLOS ONE

Dear Dr. Youn,

Thank you for submitting your manuscript to PLOS ONE. After careful consideration, we feel that it has merit but does not fully meet PLOS ONE’s publication criteria as it currently stands. Therefore, we invite you to submit a revised version of the manuscript that addresses the points raised during the review process.

We look forward to receiving your revised manuscript.

Kind regards,

Theodoros Xanthos

Academic Editor

PLOS ONE

Journal Requirements:

"This research was supported by Basic Science Research Program through the National

Research Foundation of Korea (NRF) funded by the Ministry of Education

(2018R1D1A1B07047594)"

"The funders had no role in study design, data collection and analysis, decision to

publish, or preparation of the manuscript."

4. Please ensure that you refer to Figure 1 in your text as, if accepted, production will need this reference to link the reader to the figure.

5. We note you have included a table to which you do not refer in the text of your manuscript. Please ensure that you refer to Table 5 in your text; if accepted, production will need this reference to link the reader to the Table.

Additional Editor Comments (if provided):

This article is very interresting. I would like to invite you to incorporate all 3 reviewers' comments. I really enjoyed the topic and I believe the manuscript will be significantly improved after revision.

Theodoros Xanthos MD, MSc, PhD, FAcadMed, FHEA, FCP, FESC, FERC

Professor Physiology & Pathophysiology

President

School of Medicine, European University Cyprus

Reviewers' comments:

Reviewer's Responses to Questions

**Comments to the Author**

1. Is the manuscript technically sound, and do the data support the conclusions?

Reviewer #1: Yes

Reviewer #2: Partly

Reviewer #3: Yes

2. Has the statistical analysis been performed appropriately and rigorously? 

Reviewer #1: Yes

Reviewer #2: Yes

Reviewer #3: Yes

3. Have the authors made all data underlying the findings in their manuscript fully available?

Reviewer #1: Yes

Reviewer #2: Yes

Reviewer #3: Yes

4. Is the manuscript presented in an intelligible fashion and written in standard English?

Reviewer #1: Yes

Reviewer #2: Yes

Reviewer #3: Yes

5. Review Comments to the Author

Reviewer #1: Thank you for giving me the opportunity to review your work. This is an interesting study investigating a clear knowledge gap in the field of post-resuscitation care.

There are several linguistic issues in the manuscript. It needs editing by a native English speaker. Although a reviewer is not supposed to provide secretarial support to the authors, I suggest that you type "neurological" instead of "neurologic" throughout your manuscript.

The title of your paper is confusing. Please keep it simple and catchy.

In your abstract methods state that it is a retrospective study.

Your introduction is appropriate.

Methods: Please elaborate which sedatives and paralytic agents were used in the studied patients. More details must be given about PV400 and ADC values. Not all readers are familiar with these imaging parameters.

Results: Please include a figure - flowchart of your study.

Discussion: Your discussion is weak. I suggest that you add a paragraph to briefly discuss other biomarkers for prognostication after cardiac arrest with emphasis to recent articles and in biomarkers for very early prognostication (before 48-72h). Is NSE better? Why it is better? Do we really need biomarkers?

I appreciate your list of limitations of your study.

Conclusion: I believe that you must also refer to the timing of NSE + MRI combination based on your findings.

Good luck with your revision.

GK

Reviewer #2: Very nice manuscript. Very interesting topic.

Nevertheless there are some points that need revision/clarification

1. In the abstract (method section), please clarify the type of observational study (case-control) and also the place it took place (hospital) - both in page 1 and 8

2. In introduction section (Page 9) line 3 you should write more about the different studies of TTM and use those references also (DOI: 10.1056/NEJMoa1906661, doi:10.1001/jama.2013.282173, DOI: 10.1016/j.resuscitation.2015.07.018).

3. In introduction section (Page 9) lines 4-6, be careful... The main causes of death is not pour neurological outcome but withdrawal of care due to poor neurological outcome. Re-write the sentence and use more references (eg https://doi.org/10.1016/j.resuscitation.2019.01.031)

4. Please change "neurologic" to "neurological".

5. Page 9 Lines: "However, no single test can predict neurologic outcomes with a 0% false positive rate (FPR)", please determine if it is good, bad or overall neurological outcome.

6. Page 9 Last sentence ("we hypothesized..."): Lacks rationale behind the hypothesis. How did you get the main idea for that combination and why NSE and neuroimaging and not others? Is it due to center availability or other knowledge?

7. Page 10 Methods. Why those two time points (48 and 72hrs)? Why not only 72 hrs? Explain in methods.

8. Page 11 CT section: average GWR is not the best measurement to choose. The best seems to be Basal Ganglia GWR and not aGWR (10.1016/j.resuscitation.2018.08.024). I strongly recommend to leave CT measurements out of the study cause they are misleading and may extrapolate wrong conclusions about the use of CT in CA.

9. Page 14 Results section: Please define the initial rhythm of the rest 66.1% (PEA vs Asystole). What was the cause of arrest if not cardiac?

10. Page 18 Discussion session (LIMITATIONS): Please elaborate on NSE limitations (ie "Biomarker thresholds vary with timing of measurement, reflecting their kinetics following initial release. An additional cause of inconsistency is the variability of techniques used to measure biomarkers, which can cause a significant systematic error between techniques"DOI https://doi.org/10.1186/s13054-018-2060-7, DOI: 10.1186/1756-0500-7-726). Discuss about other biomarkers (that may be more accurate), like tau-proteins from blood sampling.

The overall idea and methods (besides CT measurements) are nice.

Reviewer #3: I really enjoyed this article. Despite the retrospective design, it provides important data that will stimulate new RCTs. I agree with the authors that the ERC/ESICM prognostication algorithm needs further (and in-detail) testing.

Please add as a limitation that you only studied the target of 33 ºC and thus your findings cannot extrapolate to other TTM temperature targets.

6. PLOS authors have the option to publish the peer review history of their article (what does this mean?). If published, this will include your full peer review and any attached files.

Reviewer #1: Yes: George Karlis

Reviewer #2: No

Reviewer #3: No

---

## [Author Response · Author response to Decision Letter 0]

23 Jul 2020

We believe the peer-review process has been successful and resulted in a stronger manuscript. Our changes to the manuscript are highlighted and we have included a point-by point discussion regarding the reviewer’s concerns to the document

---

## [Decision Letter · Decision Letter 1]

17 Sep 2020

Neuron-specific enolase and neuroimaging for prognostication after cardiac arrest treated with targeted temperature management

PONE-D-20-14003R1

Dear Dr. Youn,

We’re pleased to inform you that your manuscript has been judged scientifically suitable for publication and will be formally accepted for publication once it meets all outstanding technical requirements.

Kind regards,

Andreas Schäfer

Academic Editor

PLOS ONE

Additional Editor Comments (optional):

Reviewers' comments:

Reviewer's Responses to Questions

**Comments to the Author**

1. If the authors have adequately addressed your comments raised in a previous round of review and you feel that this manuscript is now acceptable for publication, you may indicate that here to bypass the “Comments to the Author” section, enter your conflict of interest statement in the “Confidential to Editor” section, and submit your "Accept" recommendation.

Reviewer #1: All comments have been addressed

Reviewer #2: All comments have been addressed

Reviewer #3: All comments have been addressed

2. Is the manuscript technically sound, and do the data support the conclusions?

Reviewer #1: Yes

Reviewer #2: Yes

Reviewer #3: Yes

3. Has the statistical analysis been performed appropriately and rigorously? 

Reviewer #1: Yes

Reviewer #2: Yes

Reviewer #3: Yes

4. Have the authors made all data underlying the findings in their manuscript fully available?

Reviewer #1: Yes

Reviewer #2: Yes

Reviewer #3: Yes

5. Is the manuscript presented in an intelligible fashion and written in standard English?

Reviewer #1: Yes

Reviewer #2: Yes

Reviewer #3: Yes

6. Review Comments to the Author

Reviewer #1: The authors have adequately addressed my comments. I am pleased with the revision.

Reviewer #2: Very nice research field and meticulously executed idea.

One a minor revision.

Write more abouτ NSE (what is the main function of the protein and a little more details about the method of measurement). The latter is important so that the results of this observational study are reproducible.

Reviewer #3: (No Response)

7. PLOS authors have the option to publish the peer review history of their article (what does this mean?). If published, this will include your full peer review and any attached files.

Reviewer #1: **Yes: **George Karlis

Reviewer #2: No

Reviewer #3: No

---

## [Editor Report · Acceptance letter]

21 Sep 2020

PONE-D-20-14003R1

Neuron-specific enolase and neuroimaging for prognostication after cardiac arrest treated with targeted temperature management

Dear Dr. Youn:

I'm pleased to inform you that your manuscript has been deemed suitable for publication in PLOS ONE. Congratulations! Your manuscript is now with our production department.

Kind regards,

on behalf of

Prof. Dr. Andreas Schäfer 

Academic Editor

PLOS ONE